# Multi-centric Comparison of Deep Learning Models for Lesion Detection in Breast MRI

**Kai Geissler**[1]                                                KAI.GEISSLER@MEVIS.FRAUNHOFER.DE
[1] *Fraunhofer Institute for Digital Medicine MEVIS, Bremen, Germany*

**Markus Wenzel**[1,2]
[2] *Constructor University, Campus Ring 1, 28759 Bremen, Germany*

**Susanne Diekmann**[1]

**Heinrich von Busch**[3]
[3] *Digital & Automation, Diagnostic Imaging, Siemens Healthineers AG, Forchheim, Germany*

**Robert Grimm**[4]
[4] *Research & Clinical Translation, Magnetic Resonance, Siemens Healthineers AG, Erlangen, Germany*

**Hans Meine**[1]

**Editors:** Accepted for publication at MIDL 2025

## Abstract

Breast magnetic resonance imaging (MRI) is a common modality for diagnostic imaging in breast cancer, creating the need for automated image analysis to assist in early detection and diagnosis. In this study, we compared multiple deep learning-based segmentation and detection algorithms for lesion detection in dynamic contrast-enhanced (DCE) breast MRI. We used a large multicentric dataset comprising T1-weighted DCE MR images from nine clinical sites in seven countries, which encompassed diverse imaging characteristics and scanner types. We evaluated several models, including the standard nnU-Net, an adapted nnU-Net with modifications to reduce false positives, a coarse resolution version of it, the transformer-based SwinUNETR-V2 and nnDetection.

The standard nnU-Net achieved a high lesion-level sensitivity of 83.8% but produced an average of 3.334 false positives per case, which is impractical for clinical use. The adapted (coarse) nnU-Net significantly reduced false positives to 0.666 (0.397) per case with a slight decrease in sensitivity to 79.9% (75.8%). SwinUNETR-V2 achieved performance comparable to that of the adapted nnU-Net. nnDetection outperformed nnU-Net in the high-sensitivity region, but performed worse than the adapted models in the lower-sensitivity region, with respect to false positives. In conclusion, nnU-Net again provides a good baseline, but our lesion detection task motivates adaptations to reduce the number of false positives.

**Keywords:** Breast MRI, Breast Lesion Detection, Breast Lesion Segmentation, Deep Learning Breast

## 1. Introduction

Breast cancer is the most common cancer type in women worldwide and ranks fifth in terms of mortality (Sung et al., 2021). In many countries, breast MRI is part of recommendations and guidelines for the early detection of breast cancer in high-risk patients (Selamoglu and Gilbert, 2020). For women with extremely dense breast tissue, the European Society of Breast Imaging recently recommended MRI-based screening (Mann et al., 2022). This creates a high demand for automated analysis of breast magnetic resonance imaging (MRI) examinations. For diagnostic imaging, breast MRI protocols are the standard of care in many countries, although under different conditions. In this study, we compare multiple deep learning-based segmentation and detection algorithms to detect lesions on dynamic contrast enhanced (DCE) MRI. These automated detection approaches may help radiologists perform the task more accurately and efficiently.

Many studies have already investigated the task of tumor detection / segmentation in breast MRI. Dalmış et al. (2018) developed a lesion detection model on 385 MRI scans based on the search for lesion proposals and subsequent classification. They achieved an average sensitivity of 0.6429 when averaging over operating points from 0.125 to 8 false positives. Zhang et al. (2018) used multiple stages of U-Nets to train a lesion segmentation model on 285 patients. They achieved a Dice score of 0.7176. Counting a lesion overlap of at least 0.50 as true positive, they achieved a sensitivity of 0.9286 with a precision of 0.6783. Zhu et al. (2022) developed a V-Net-based segmentation model on 2,823 patients from 2 clinical sites, achieving a Dice score of 0.860. Zhang et al. (2023) developed a lesion segmentation model on 2,190 patients from 7 clinical sites who achieved a Dice score of 0.724. Park et al. (2024) developed a lesion segmentation model on examinations of 736 women from a single institution. They used manual regions of interest with a combination of manual and automatic correction and achieved a Dice score of 0.75 when evaluated over the entire volume and a Dice score of 0.89 when evaluated per tumor. The majority of the work is based on lesion segmentation models similar to those mentioned so far. A study using detection architectures (Zhang et al., 2022) trained a Mask-RCNN for lesion detection on 241 patients. They achieved a slice level sensitivity of 0.81 at 2 false positives per image and a slice level Dice score of 0.79.

In this study, we train models for lesion detection based on nnDetection (Baumgartner et al., 2021), SwinUNETR-V2 (He et al., 2023), nnU-Net (Isensee et al., 2021), and adapted versions of the latter. They are trained and evaluated on a large multi-centric dataset.

## 2. Methods

### 2.1. Data

We used a multi-centric dataset of T1-weighted dynamic contrast-enhanced MR images of the female breast collected from 2,751 patients patients at 9 clinical sites in seven different countries spanning Eastern and Western Europe, North America, and Asia. Data from five sites were used to train breast segmentation models, while data from all sites were used to train lesion segmentation models. The data cover a diverse set of imaging characteristics, such as different fields of view, resolutions, voxel sizes, fat suppression (FS) settings, and MR scanners. The MR scanners include models from Siemens Healthineers (Forchheim,

Germany), GE HealthCare (Chicago, United States) and Philips (Best, The Netherlands). An overview of the most important image characteristics and the number of images and masks for all sites is shown in Table 1. Additional information on scanner vendors, MR sequence parameters and patient characteristics is shown in Table 4 in the appendix. A visual impression of the different image characteristics is provided in Figure 3 in the appendix.

The dataset identified with "Site 8" in Table 1 is a subset of the ACRIN-6698 dataset (Newitt et al., 2021) for which we manually corrected the tumor segmentation masks, while the remaining data are a proprietary collection.

215 breast masks were iteratively created by a radiologist, a radiological technologist, and a research scientist. The cases were randomly selected. A deep learning model was used to create segmentation proposals that were corrected by the annotators. After annotating a certain number of images, the deep learning model was retrained from scratch to provide better segmentation proposals. The first models were trained using breast masks created by classical image processing. The masks were created on the data from sites 1–5 while the data from sites 6–9 have no breast masks.

To train the lesion segmentation model, a radiologist and 3 radiological technologists manually segmented 2,318 lesion masks on data from all available sites.

We divided all cases with masks into training, validation, and test groups using a 60-20-20 percent split for training the segmentation and detection models. The annotation was performed iteratively and the assignment of cases was done randomly. Since the assignment algorithm did not stratify the sampling according to the site, the split proportions for each site differ slightly.

## 2.2. Segmentation and Detection Models

Our lesion detection pipeline acts in two steps: First, a region of interest around the breast region is cropped from the image. Then we apply a lesion segmentation (or detection) algorithm to detect lesions in the breast region. All of the following methods work directly on 3D MRI volumes using 3D deep learning architectures.

To segment the breast region of interest, we train a 3D U-Net which uses 5 levels with only 6 base filters. We use an AdamW optimizer, a batch size of 2 and spatial dropout with a drop rate of 0.2 on each level, except for the highest level in the encoder and the decoder part of the U-Net. A learning rate of 0.0001 is used with a cosine annealing learning rate scheduler and instance normalization. The input images are resampled to a voxel size of $2 \times 2 \times 2$ mm$^3$ and only the T1 precontrast image is used as input. The model hyperparameters and resampling settings are chosen by manual tuning to achieve a practical trade-off between inference time and segmentation performance. For data augmentation, we use the batch generators library (Isensee et al., 2020).

As baselines for our lesion detection, we train a low-resolution nnU-Net (Isensee et al., 2021) and an nnDetection (Baumgartner et al., 2021) model. Both are self-configuring deep learning models which use a mix of fixed parameters and heuristics to find well fitting training configurations for the given task. While nnU-Net produces a segmentation model with dense voxel-wise output masks, nnDetection trains a detection model which provides output boxes with classification scores for each detected box. Both models use the precontrast image and the postcontrast image which is closest to 60 seconds after the first postcontrast

Table 1: Data description per clinical site. *: Mean ± Standard Deviation

|  | Site 1 | Site 2 | Site 3 |
|---|---|---|---|
| Patients | 930 | 44 | 60 |
| Studies | 936 | 44 | 60 |
| Breast Masks (patients) | 31 (31) | 35 (35) | 30 (30) |
| Lesion Masks (patients) | 669 (452) | 72 (42) | 94 (60) |
| Fat Suppression | Yes / No | Yes / No | Yes |
| Voxel Size (in-plane)* [mm] | $0.44 \pm 0.0$ | $0.87 \pm 0.07$ | $1.02 \pm 0.01$ |
| Slice Thickness* [mm] | $1.8 \pm 0.02$ | $1.5 \pm 0.0$ | $1.2 \pm 0.0$ |
| Resolution (in-plane)* | $896.0 \pm 0.0$ | $432.0 \pm 28.0$ | $352.0 \pm 0.0$ |
| Number of Slices* | $112.4 \pm 3.4$ | $116.7 \pm 4.0$ | $160.8 \pm 3.5$ |
| Country | Austria | Germany | Poland |
|  | **Site 4** | **Site 5** | **Site 6** |
| Patients | 960 | 230 | 194 |
| Studies | 1159 | 231 | 194 |
| Breast Masks (patients) | 32 (32) | 19 (19) | 0 (0) |
| Lesion Masks (patients) | 613 (479) | 253 (230) | 254 (193) |
| Fat Suppression | No | Yes | Yes |
| Voxel Size (in-plane)* [mm] | $0.75 \pm 0.02$ | $0.8 \pm 0.12$ | $0.89 \pm 0.07$ |
| Slice Thickness* [mm] | $2.38 \pm 0.24$ | $1.06 \pm 0.34$ | $1.0 \pm 0.03$ |
| Resolution (in-plane)* | $512.0 \pm 0.0$ | $411.8 \pm 137.3$ | $372.9 \pm 23.7$ |
| Number of Slices* | $46.0 \pm 0.4$ | $187.3 \pm 32.6$ | $147.6 \pm 7.8$ |
| Country | Germany | Japan | Japan |
|  | **Site 7** | **Site 8** | **Site 9** |
| Patients | 36 | 194 | 103 |
| Studies | 36 | 194 | 104 |
| Breast Masks (patients) | 1 (1) | 0 (0) | 0 (0) |
| Lesion Masks (patients) | 37 (24) | 234 (132) | 92 (84) |
| Fat Suppression | Yes | Yes | Yes |
| Voxel Size (in-plane)* [mm] | $0.75 \pm 0.07$ | $0.67 \pm 0.07$ | $0.86 \pm 0.09$ |
| Slice Thickness* [mm] | $1.44 \pm 0.15$ | $1.8 \pm 0.47$ | $1.5 \pm 0.0$ |
| Resolution (in-plane)* | $434.8 \pm 33.0$ | $509.3 \pm 46.4$ | $442.0 \pm 45.5$ |
| Number of Slices* | $116.2 \pm 11.6$ | $113.8 \pm 33.8$ | $154.2 \pm 8.9$ |
| Country | India | USA | Israel |

image, provided as two input channels. The postcontrast timepoint is selected based on the heuristic that contrast agent is applied roughly 60 seconds before the first post-contrast image, as the timing of contrast agent application was lost during data anonymization.

We compare these models with our reimplementation of the nnU-net training setup, of which we adapt some parts. First, we switch from a channel-wise normalization of the input images to one that uses the same normalization parameters for all channels. The reason is that in contrast-enhanced images most tissues (such as bones, muscles, and fat) do not change their intensities, while the heart, vessels, and breast tissue enhance after contrast agent application. Second, we change the patch sampling to sample all sites equally often, to reduce the bias towards the larger sites during patch sampling. Patches with and without foreground (lesion) voxels are sampled equally often for each site. For sites for which the data contains cases without lesions (i.e. BI-RADS 1 cases or BI-RADS 2 cases without enhancing lesions), the background patches are sampled only from these cases. We do this because in some cases with lesions, only the index lesion has been marked and not all the lesions present in the case, as this is the clinically most relevant lesion. Thus, we try to avoid including these false negative lesions in the training data. The model with these two modifications is called Adapted nnU-Net in the following.

The original low-resolution nnU-Net model still acts on a relatively small voxel size of $0.74 \times 0.74 \times 1.8 \, \text{mm}^3$. As this leads to long inference times, we train an additional model using a coarse voxel size of $1 \times 1 \times 2 \, \text{mm}^3$. This model will be called Coarse nnU-Net. While we change the data preprocessing, we use the same neural network architecture for all nnU-Net models.

Transformer-based segmentation models have been shown to provide superior performance over purely convolutional segmentation models in several studies (Xiao et al., 2023). Therefore, we also train a SwinUNETR-V2 (He et al., 2023) which uses a shifted window transformer as the encoder and convolutional layers for its decoder part. We use the same training pipeline as our adapted nnU-Net model. The learning rate is changed from 0.01 to 0.0004.

A summary of data pre-processing for all models is provided in Table 5 in the appendix.

## 3. Results

The lesion-level results on our test data are shown in Table 2. The nnU-Net model achieves a lesion-level sensitivity of 0.838 with an average of 3.334 false positives per case. The adapted (coarse) nnU-Net model achieves a sensitivity of 0.799 (0.758) with an average amount of false positives of 0.666 (0.397). SwinUNETR-V2 achieves a sensitivity of 0.792 with an average false positive amount of 0.722. We evaluated nnDetection at 3 different thresholds on its confidence scores: 0.50, 0.90 and 0.98. It achieves a sensitivity of 0.942, 0.846 and 0.699, respectively, while finding 3.538, 1.685 and 0.920 false positives on average.

With respect to per-lesion segmentation performance, all segmentation models perform very similar, ranging from 0.742 (Coarse nnU-Net) to 0.752 (nnU-Net) in terms of per-lesion bounding box intersection over union. The Dice per lesion ranges from 0.584 (coarse nnU-Net) to 0.599 (nnU-Net). nnDetection achieves smaller bounding box overlaps ranging from 0.409 to 0.507, depending on the confidence threshold. The qualitative results of the lesion segmentation / detection models are shown in Figure 1.

Table 2: Lesion level evaluation on test set. The mean and 90% confidence interval are shown (computed via bootstrapping).
(AFP: Average False Positives Per Case, BBIoU: Bounding Box Intersection over Union per Lesion, pL Dice: Per Lesion Dice)

| Model | Sensitivity | AFP | BBIoU | pL Dice |
|---|---|---|---|---|
| nnU-Net | $0.838_{[0.809,0.867]}$ | $3.334_{[3.168,3.501]}$ | $0.752_{[0.737,0.768]}$ | $0.599_{[0.580,0.619]}$ |
| Adapted nnU-Net | $0.799_{[0.769,0.829]}$ | $0.666_{[0.597,0.739]}$ | $0.746_{[0.729,0.763]}$ | $0.594_{[0.573,0.614]}$ |
| Coarse nnU-Net | $0.758_{[0.728,0.788]}$ | $0.397_{[0.346,0.447]}$ | $0.742_{[0.724,0.759]}$ | $0.584_{[0.564,0.604]}$ |
| SwinUNETR-V2 | $0.792_{[0.761,0.823]}$ | $0.722_{[0.647,0.802]}$ | $0.742_{[0.725,0.759]}$ | $0.592_{[0.572,0.612]}$ |
| nnDetection@0.50 | $0.942_{[0.928,0.956]}$ | $3.538_{[3.484,3.592]}$ | $0.409_{[0.394,0.425]}$ | - |
| nnDetection@0.90 | $0.846_{[0.824,0.869]}$ | $1.685_{[1.638,1.730]}$ | $0.474_{[0.458,0.491]}$ | - |
| nnDetection@0.98 | $0.699_{[0.669,0.727]}$ | $0.920_{[0.880,0.959]}$ | $0.507_{[0.489,0.525]}$ | - |

Table 3: Case level evaluation on internal dataset. The mean and 90% confidence interval are shown (computed via bootstrapping).
(FPR: False Positive Rate)

| Model | Sensitivity | FPR |
|---|---|---|
| nnU-Net | $0.959_{[0.941,0.976]}$ | $0.974_{[0.955,0.991]}$ |
| Adapted nnU-Net | $0.918_{[0.894,0.941]}$ | $0.364_{[0.311,0.417]}$ |
| Coarse nnU-Net | $0.885_{[0.858,0.912]}$ | $0.281_{[0.233,0.329]}$ |
| SwinUNETR-V2 | $0.902_{[0.876,0.927]}$ | $0.355_{[0.302,0.408]}$ |
| nnDetection@0.50 | $0.939_{[0.920,0.956]}$ | $0.727_{[0.684,0.769]}$ |
| nnDetection@0.90 | $0.860_{[0.834,0.886]}$ | $0.491_{[0.443,0.540]}$ |
| nnDetection@0.98 | $0.740_{[0.707,0.774]}$ | $0.225_{[0.186,0.266]}$ |

We also compute the detection performance on the case level. The resulting metrics are shown in Table 3. nnU-Net achieves a case-level sensitivity of 0.959 with a false positive rate of 0.974. Our adapted (coarse) nnU-Net achieves a sensitivity of 0.918 (0.885) with a false positive rate of 0.364 (0.281). SwinUNETR-V2 performs similarly to the adapted nnU-Net, having a sensitivity of 0.902 at a false positive rate of 0.355. When evaluating nnDetection at a confidence threshold of 0.5 (0.98), it achieves a case-level sensitivity of 0.939 (0.740) at a false positive rate of 0.727 (0.225).

Figure 2 shows the lesion level sensitivity per site and models, once clustered by models and once by site. The respective tables with exact metrics per model and per site can be found in Appendix D. We can observe that there is a certain variation in terms of sensitivity and average false positives per site. This can be explained by the widely different image characteristics and patient cohorts that originate from different sites. Still, we observe that more sensitive (/specific) models tend to be more sensitive (/specific) across all sites.

Analyzing whether the models learned any detrimental biases from the data will be the subject of future work.

## 4. Discussion

In this study, we compared various deep learning-based segmentation and detection algorithms for lesion detection in dynamic contrast-enhanced breast MRI using a large, multi-centric dataset. Although the standard nnU-Net achieved a high lesion-level sensitivity of 83.8%, it produced an impractically high average of 3.334 false positives per case, limiting its clinical utility.

By adapting nnU-Net with modifications to input normalization and sampling strategies, we significantly reduced the average false positives to 0.666 per case, with only a slight decrease in sensitivity to 79.9%. The coarse resolution version further reduced false positives to 0.397 per case but saw a modest sensitivity drop to 75.8%. We hypothesize that the reduction of the coarse nnU-Net is due to the lower resolution filtering out fine-grained details that may contribute to false positive predictions. High-resolution images provide detailed anatomical structures, which can sometimes lead the model to misinterpret normal variations in tissue or imaging artifacts as lesions. By reducing the resolution, these fine details are "smoothed out", allowing the model to focus on larger, more salient features that are characteristic of true lesions. These results indicate that strategic adjustments to the nnU-Net can enhance its practicality for clinical applications by reducing false positives while maintaining acceptable sensitivity.

The transformer-based SwinUNETR-V2 performed comparably to the adapted nnU-Net, with a sensitivity of 79.2% and 0.722 false positives per case, suggesting that transformer architectures are viable alternatives to traditional convolutional models for this task. Yet in our experiments it does not improve over the solely convolutional architectures.

nnDetection achieved a higher lesion-level sensitivity than nnU-Net at a confidence threshold of 0.50 while producing a similar amount of false positives. At a threshold of 0.90 it matches the lesion-level sensitivity of nnU-Net while producing only half as many false positives. At a confidence threshold of 0.98 its sensitivity drops below the one of adapted and coarse nnU-Net while still producing more false positives, indicating that it is most useful in the high-sensitivity region. In general, the confidence scores associated to predictions of detection models are beneficial when models are employed in clinical practice. They allow their users to select a sensitivity-specificity trade-off that fits their needs, as different image characteristics and patient cohorts will usually change the model behavior at least slightly.

Our findings highlight the challenge of balancing sensitivity and specificity in automated lesion detection, as higher sensitivity tended to produce more false positives, and this trend was consistent across the diverse imaging settings from different clinical sites. The adapted nnU-Net models offer a good balance, reducing false positives to a clinically acceptable level without substantially compromising sensitivity. The robustness of these models across diverse imaging settings in our multi-centric dataset further underscores their potential clinical applicability.

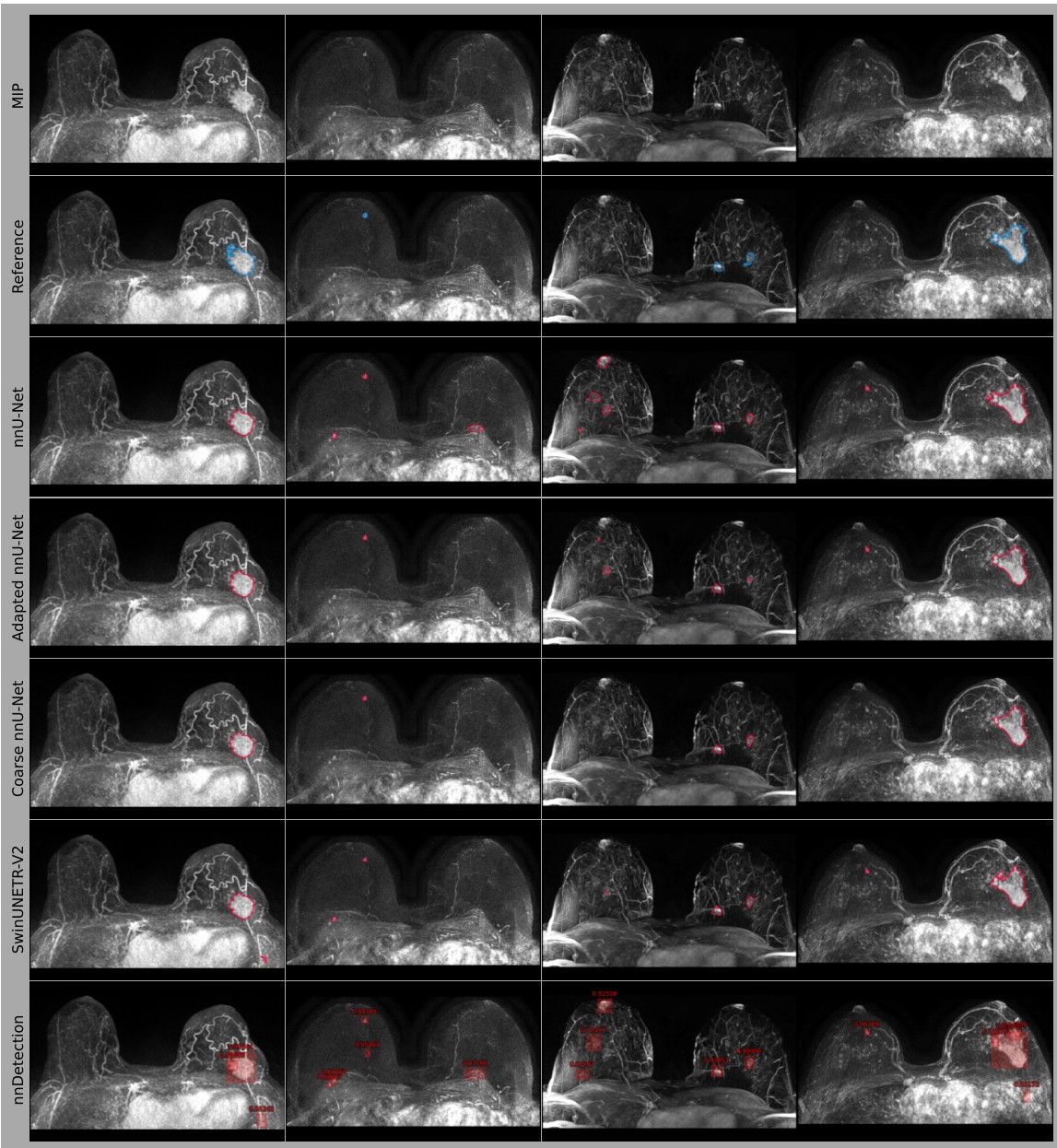

Figure 1: 2D projections of the segmentation masks / detected bounding boxes with confidences are overlayed on top of the maximum-intensity-projection (MIP) of the difference image. (All methods work on 3D volumes using 3D models. MIPs are shown to provide a better overview of the cases.)

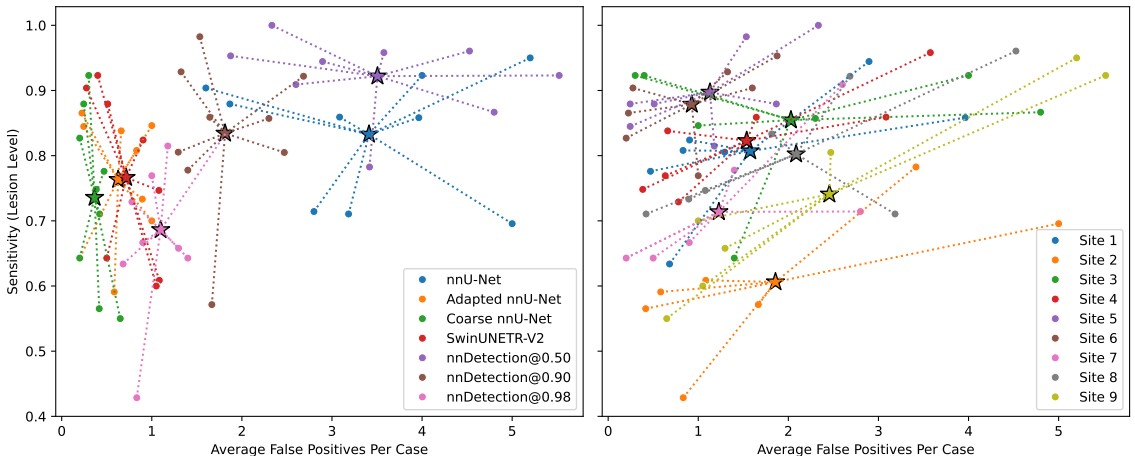

Figure 2: Per site lesion-level results, grouped by model (left) and by site (right). It is visible that more sensitive or specific models are so across different sites, as they cluster together to a certain extent. Also site 2 appears to be an outlier where all models achieve a very low sensitivity. The star marks the mean for each model (left) or each site (right).

## 5. Conclusion

This study compared segmentation and detection algorithms for lesion detection in breast MRI using a large multi-centric dataset. The standard nnU-Net, despite its high sensitivity, produced too many false positives for practical clinical use. By adapting nnU-Net, we significantly reduced false positives to a clinically acceptable level with a small loss of sensitivity. The coarse resolution nnU-Net further decreased false positives at the cost of an additional sensitivity reduction. SwinUNETR-V2 performed similarly to the adapted nnU-Net, indicating that transformer-based models are suitable alternatives but do not provide a performance benefit on this dataset. nnDetection outperformed nnU-Net in the high-sensitivity region but performed worse than the adapted nnU-Net models in the lower-sensitivity region.

In summary, with appropriate modifications, segmentation-based deep learning models such as the adapted nnU-Net provide a favorable balance between sensitivity and the amount of false positives for lesion detection in breast MRI. These models are a good basis for further developing computer-assisted diagnostic support for more efficient and accurate breast MRI reading.

## Acknowledgments

We thank Christiane Engel, Sophia Winkler and Andrea Koller for their efforts in creating the segmentation masks. We also thank our clinical partners for providing the data used in this study: Dieter Szolar (Diagnostikum Graz, Graz, Austria), Sabine Ohlmeyer (Uni-

versitätsklinikum Erlangen, Germany), Edyta Szurowska (Second Radiology Department, Medical University of Gdansk, Gdansk, Poland), Uwe Fischer (Diagnostic Breast Center Göttingen, Göttingen, Germany), Nachiko Uchiyama (Department of Radiology, Nippon Medical School Hospital, Tokyo, Japan), Kazuki Oyama (Department of Radiology, Shinshu University School of Medicine, Matsumoto, Japan), Noemi Schmidt (Radiology and Nuclear Medicine, University Hospital Basel, Basel, Switzerland), Pratiksha Yadav (Dr. D. Y. Patil Hospital, Pune, India) and Miri Sklair-Levy (Sheba Medical Center, Ramat Gan, Israel).

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

# Appendix A. Additional Data Information

Table 4: Additional data description of patient collectives, MRI scanners and MRI sequences per clinical site. *: Mean ± Standard Deviation, SHS: Siemens, Ph: Philips

|  | Site 1 | Site 2 | Site 3 | Site 4 | Site 5 |
|---|---|---|---|---|---|
| Age* [years] | $52.0 \pm 13.0$ | $47.8 \pm 14.3$ | $49.3 \pm 10.8$ | $55.5 \pm 10.7$ | $53.2 \pm 12.1$ |
| TR* [ms] | $4.89 \pm 0.0$ | $6.0 \pm 0.41$ | $3.35 \pm 0.0$ | $8.45 \pm 0.07$ | $3.51 \pm 0.57$ |
| TE* [ms] | $1.8 \pm 0.01$ | $3.2 \pm 1.09$ | $1.11 \pm 0.0$ | $4.05 \pm 0.04$ | $1.5 \pm 0.36$ |
| Field Strength [T] | 3 | 1.5, 3 | 1.5 | 1.5 | 1.5, 3 |
| Manufacturer | SHS | SHS | SHS | GE | SHS,Ph,GE |
| Flip Angle [°] | 9, 10 | 8-10 | 10 | 10 | 8, 10-15 |
| BI-RADS | 1-6 | 2, 6 | 2-6 | 1-5 | 3-5 |
|  | Site 6 | Site 7 | Site 8 | Site 9 |  |
| Age* [years] | $58.9 \pm 13.5$ | $44.9 \pm 13.3$ | - | $55.3 \pm 13.1$ |  |
| TR* [ms] | $4.03 \pm 0.61$ | $4.65 \pm 0.23$ | $6.79 \pm 1.59$ | $5.19 \pm 0.24$ |  |
| TE* [ms] | $1.58 \pm 0.38$ | $1.84 \pm 0.25$ | $3.41 \pm 1.06$ | $1.82 \pm 0.54$ |  |
| Field Strength [T] | 1.5, 3 | 1.5, 3 | 1.5, 3 | 3 |  |
| Manufacturer | SHS | SHS | SHS,Ph,GE | SHS |  |
| Flip Angle [°] | 9, 10 | 10 | 10-17 | 10 |  |
| BI-RADS | 1, 3-5 | 1-6 | 6 | 2-4, 6 |  |

## Appendix B. Example Images

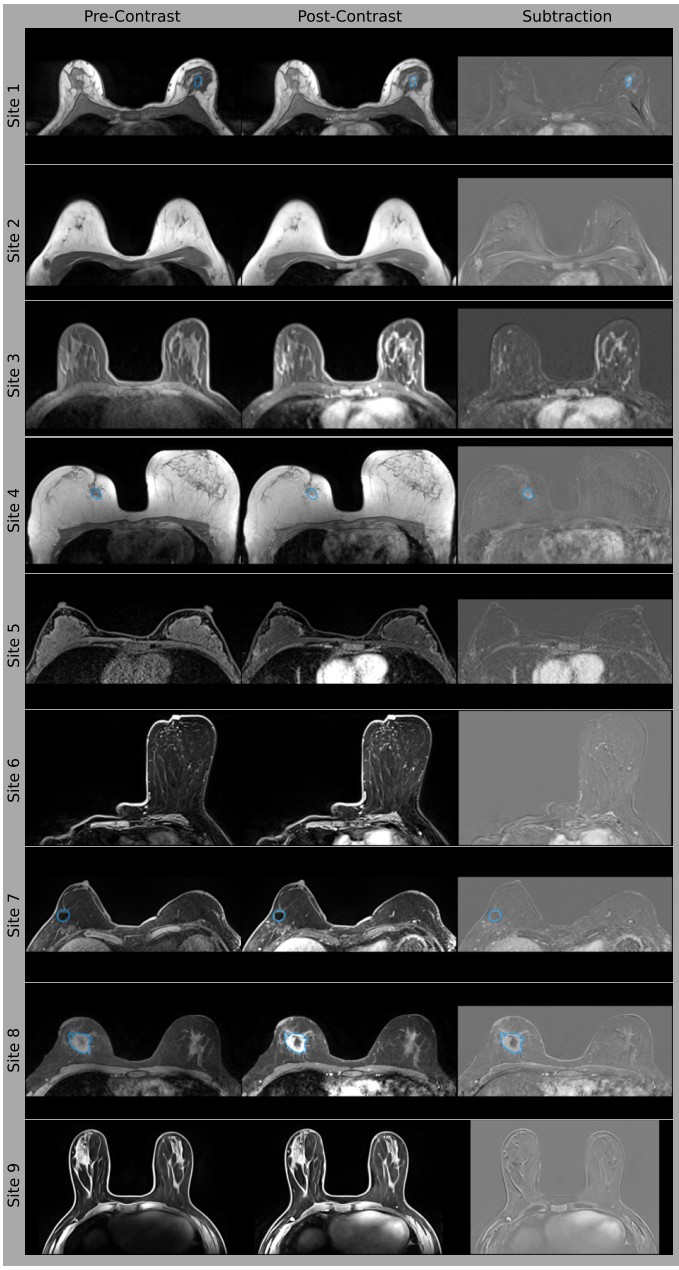

Figure 3: Central slices of exemplary MRI volumes for each site. Slices are cropped to breast region. Lesion masks are contoured in blue. Pre-Constrast: MR image before contrast injection. Post-Contrast: MR image after contrast injection. Subtraction: Post-Contrast minus Pre-Contrast.

## Appendix C. Overview of Data Preprocessing

Table 5: Data preprocessing for the deep learning models.

|                         | nnU-Net | Adapted nnU-Net | Coarse nnU-Net |
|-------------------------|---------|-----------------|----------------|
| Voxel Size [mm]         | $0.74 \times 0.74 \times 1.8$ | $0.74 \times 0.74 \times 1.8$ | $1 \times 1 \times 2$ |
| Patch Size              | $288 \times 128 \times 56$ | $288 \times 128 \times 56$ | $288 \times 128 \times 56$ |
| Intensity Normalization | per channel z-score | percentile mapping | percentile mapping |
| Patch Sampling          | nnU-Net strategy | Our (see Sec. 2.2) | Our (see Sec. 2.2) |
| Input Region            | breast crop | breast crop | breast crop |
| Implementation          | nnU-Net | Our | Our |

|                         | SwinUNETR-V2 | nnDetection | |
|-------------------------|--------------|-------------|---|
| Voxel Size [mm]         | $0.74 \times 0.74 \times 1.8$ | $0.74 \times 0.74 \times 1.8$ | |
| Patch Size              | $288 \times 128 \times 64$ | $224 \times 128 \times 64$ | |
| Intensity Normalization | percentile mapping | per channel z-score | |
| Patch Sampling          | Our (see Sec. 2.2) | nnDetection strategy | |
| Input Region            | breast crop | breast crop | |
| Implementation          | Our/MONAI[*] | nnDetection | |

[*] We use the MONAI (Cardoso et al., 2022) implementation of the SwinUNETR-v2 architecture with our own data processing and model training pipeline.

## Appendix D. Metrics for Each Site Individually

For all tables in this section, we use the following abbreviations: Average FP: Average False Positives Per Case, BBIoU: Bounding Box Intersection over Union per Lesion.

Table 6: Metrics per Site for nnU-Net

| Site | Sensitivity | Average FP | BBIoU | per Lesion Dice |
|------|-------------|------------|-------|-----------------|
| Site 1 | 0.858 | 3.964 | 0.617 | 0.748 |
| Site 2 | 0.696 | 5.000 | 0.424 | 0.694 |
| Site 3 | 0.923 | 4.000 | 0.831 | 0.888 |
| Site 4 | 0.859 | 3.085 | 0.581 | 0.748 |
| Site 5 | 0.879 | 1.867 | 0.664 | 0.814 |
| Site 6 | 0.904 | 1.600 | 0.557 | 0.715 |
| Site 7 | 0.714 | 2.800 | 0.553 | 0.699 |
| Site 8 | 0.711 | 3.184 | 0.638 | 0.753 |
| Site 9 | 0.950 | 5.200 | 0.460 | 0.706 |

Table 7: Metrics per Site for Adapted nnU-Net

| Site | Sensitivity | Average FP | BBIoU | per Lesion Dice |
|------|-------------|------------|-------|-----------------|
| Site 1 | 0.808 | 0.831 | 0.617 | 0.764 |
| Site 2 | 0.591 | 0.583 | 0.412 | 0.650 |
| Site 3 | 0.846 | 1.000 | 0.788 | 0.896 |
| Site 4 | 0.838 | 0.661 | 0.543 | 0.722 |
| Site 5 | 0.845 | 0.244 | 0.677 | 0.816 |
| Site 6 | 0.865 | 0.225 | 0.590 | 0.720 |
| Site 7 | 0.643 | 0.200 | 0.626 | 0.722 |
| Site 8 | 0.733 | 0.895 | 0.626 | 0.741 |
| Site 9 | 0.700 | 1.000 | 0.446 | 0.658 |

Table 8: Metrics per Site for Coarse nnU-Net

| Site | Sensitivity | Average FP | BBIoU | per Lesion Dice |
|------|-------------|------------|-------|-----------------|
| Site 1 | 0.776 | 0.470 | 0.589 | 0.749 |
| Site 2 | 0.565 | 0.417 | 0.455 | 0.650 |
| Site 3 | 0.923 | 0.300 | 0.703 | 0.832 |
| Site 4 | 0.748 | 0.384 | 0.544 | 0.728 |
| Site 5 | 0.879 | 0.244 | 0.660 | 0.783 |
| Site 6 | 0.827 | 0.200 | 0.600 | 0.747 |
| Site 7 | 0.643 | 0.200 | 0.605 | 0.729 |
| Site 8 | 0.711 | 0.421 | 0.606 | 0.741 |
| Site 9 | 0.550 | 0.650 | 0.440 | 0.627 |

Table 9: Metrics per Site for SwinUNETR-V2

| Site | Sensitivity | Average FP | BBIoU | per Lesion Dice |
|------|-------------|------------|-------|-----------------|
| Site 1 | 0.824 | 0.904 | 0.596 | 0.735 |
| Site 2 | 0.609 | 1.083 | 0.422 | 0.674 |
| Site 3 | 0.923 | 0.400 | 0.748 | 0.862 |
| Site 4 | 0.769 | 0.634 | 0.551 | 0.725 |
| Site 5 | 0.879 | 0.511 | 0.694 | 0.816 |
| Site 6 | 0.904 | 0.275 | 0.591 | 0.741 |
| Site 7 | 0.643 | 0.500 | 0.484 | 0.602 |
| Site 8 | 0.747 | 1.079 | 0.599 | 0.725 |
| Site 9 | 0.600 | 1.050 | 0.556 | 0.751 |

Table 10: Metrics per Site for nnDetection@0.50

| Site | Sensitivity | Average FP | BBIoU |
|---|---|---|---|
| Site 1 | 0.944 | 2.925 | 0.392 |
| Site 2 | 0.783 | 3.333 | 0.420 |
| Site 3 | 0.867 | 5.200 | 0.507 |
| Site 4 | 0.964 | 3.572 | 0.366 |
| Site 5 | 1.000 | 2.622 | 0.538 |
| Site 6 | 0.952 | 1.925 | 0.440 |
| Site 7 | 0.923 | 2.700 | 0.433 |
| Site 8 | 0.961 | 4.763 | 0.317 |
| Site 9 | 0.933 | 5.855 | 0.445 |

Table 11: Metrics per Site for nnDetection@0.90

| Site | Sensitivity | Average FP | BBIoU |
|---|---|---|---|
| Site 1 | 0.805 | 1.308 | 0.474 |
| Site 2 | 0.571 | 1.667 | 0.464 |
| Site 3 | 0.857 | 2.400 | 0.501 |
| Site 4 | 0.872 | 1.619 | 0.435 |
| Site 5 | 0.948 | 1.711 | 0.564 |
| Site 6 | 0.909 | 1.350 | 0.502 |
| Site 7 | 0.900 | 1.600 | 0.518 |
| Site 8 | 0.919 | 2.658 | 0.352 |
| Site 9 | 0.815 | 2.623 | 0.508 |

Table 12: Metrics per Site for nnDetection@0.98

| Site | Sensitivity | Average FP | BBIoU |
|---|---|---|---|
| Site 1 | 0.634 | 0.677 | 0.493 |
| Site 2 | 0.429 | 0.833 | 0.486 |
| Site 3 | 0.714 | 1.500 | 0.542 |
| Site 4 | 0.766 | 0.877 | 0.446 |
| Site 5 | 0.852 | 1.244 | 0.619 |
| Site 6 | 0.827 | 1.075 | 0.537 |
| Site 7 | 0.667 | 1.000 | 0.542 |
| Site 8 | 0.784 | 1.684 | 0.424 |
| Site 9 | 0.673 | 1.348 | 0.543 |

