# OpenReview forum: "Multi-centric Comparison of Deep Learning Models for Lesion Detection in Breast MRI"
_MIDL.io/2025/Conference — MIDL 2025 Poster_

### Official Review · Reviewer_UGYw · 2025-02-17

**Confidence:** 5
**Preliminary Rating:** 1
**Recommendation:** Poster
**Final Rating:** 1

**Summary:**

The paper uses multiple off-the-shelf deep learning methods including nnUnet, SwinUNETR to segment and detect breast cancer lesions and evaluate on multi-center datasets. Analysis of these models on multiple center datasets is good since MRI can be highly variable across institutions. However, the novelty of the work is highly limited. Also, given so many center datasets were used, it would have been more interesting to see how the models performed on the individual datasets and under what conditions the models work and where they fail. Although the use of data from 8 different centers is nice, the analysis falls short of depth in terms of how to assess the generalizable capabilities of the various methods.

**Strengths:**

* Use of 8 different center datasets.
* Paper also presents characteristics of the data from the 8 centers
* Provides a reasonably good description of the various methods and explains how the training, validation, and testing was done.

**Weaknesses:**

* Methodological novelty is highly limited. Analysis involves the use of off-the-shelf methods.
* Rigor of analysis is also limited as there is no description of how the models fare across the individual center datasets.
* It would also be interesting to see how the model if trained with data from one center extend across to other centers.

**Detailed Comments:**

The paper uses multiple off-the-shelf deep learning methods including nnUnet, SwinUNETR to segment and detect breast cancer lesions and evaluate on multi-center datasets. Analysis of these models on multiple center datasets is good since MRI can be highly variable across institutions. However, the novelty of the work is highly limited. Also, given so many center datasets were used, it would have been more interesting to see how the models performed on the individual datasets and under what conditions the models work and where they fail. Although the use of data from 8 different centers is nice, the analysis falls short of depth in terms of how to assess the generalizable capabilities of the various methods.

Please see list of weaknesses.

**Justification Of The Final Rating:**

Thank you for including the additional site-specific metrics for each model. However, as the key novelty of this work is the analysis of generalizability of the various models to the different datasets, lack of experiments to really understand that issue, makes the paper very limited in applicability as no new insights are gained about these methods. For these reasons, I will keep my score the same.

**Justification Of The Preliminary Rating:**

The methodological and conceptual novelty is very limited. The paper mostly uses off-the-shelf methods for the analysis. For this reason, I don't think the paper in the current form is of sufficient interest to the readership.

**Questions To Address In The Rebuttal:**

* Methodological novelty is highly limited. Please clarify what the novelty of this work is.
* Rigor of analysis is also limited as there is no description of how the models fare across the individual center datasets. Please show how the methods generalize, where they work well and where and why they fail across the different centers.
* It would also be interesting to see how the model if trained with data from one center extend across to other centers.

**Special Issue:**

No

---

> ### Author Response · Authors · 2025-03-07
>
> Dear reviewer, we thank you for your valuable comments and feedback and addressed it in our draft as follows:
>
> > Methodological novelty is highly limited. Please clarify what the novelty of this work is.
>
> We agree with you, that the methodological novelty of our work is very limited. However, this study was desinged as an application study, which evaluated several established state-of-the-art algorithms for breast lesion detection on a new, large, multi-centric dataset. Additionally we highlight the clinical challenge of trading off sensitivity vs. specificity. Therefore we feel that this study is clearly within the scope of the MIDL conference and of interest for its community.
>
> ---
>
> > Rigor of analysis is also limited as there is no description of how the models fare across the individual center datasets. Please show how the methods generalize, where they work well and where and why they fail across the different centers.
>
> We added Tables 5-11 to the appendix, which show the per site metrics for each model. As several variables change at the same time for the different sites (like patient cohorts and multiple imaging characterstics), it is hard, if not infeasible, to disentangle these effects for an analysis of why the models fail across different centers. Given the limited time for the rebuttal it was not possible to perform this extensive analysis.
>
> ---
>
> > It would also be interesting to see how the model if trained with data from one center extend across to other centers.
>
> We agree that this would be a valuable experiment, to be able to better judge the generalization capabilites of the models. However training each model takes 1-2 weeks, so it was not possible to perform this experiment for the rebuttal.

---

> > ### Comment · Reviewer_UGYw · 2025-03-13
> > **Somewhat addresses the concerns but the novelty is still limited**
> >
> > Thank you for including the additional site-specific metrics for each model. However, as the key novelty of this work is the analysis of generalizability of the various models to the different datasets, lack of experiments to really understand that issue, makes the paper very limited in applicability as no new insights are gained about these methods. For these reasons, I will keep my score the same.

---

### Official Review · Reviewer_DVGD · 2025-02-19

**Confidence:** 4
**Preliminary Rating:** 5
**Recommendation:** Oral
**Final Rating:** 5

**Summary:**

The authors present an extensive experimental study of lesion detection and segmentation in T1 breast MRI, over a wide range of datasets (nine clinical sites and seven countries) and five different detection or segmentation models. In addition to providing a general benchmark, their results include various interesting and useful findings, including nnU-Net's strong performance yet high false positive rate (and the general tradeoff between sensitivity and FP rate).

**Strengths:**

(1) The experiments are conducted on a wide range of datasets from nine clinical sites, from seven countries. Table 1 shows that this results in a wide range of imaging settings, e.g. slice thickness/number, TE, TR, flip angle, manufacturer, etc, altogether resulting in a comprehensive benchmark for this task.
(2) The experimental design, which is of high importance for a paper like this, is solid. The dataset/split creation is good, as is their scan pre-processing which is well-justified. Their evaluation pipeline and metrics are reasonable, and the addition of confidence intervals is demonstrates the robustness of their results.
(3) The authors experiment on strong, state-of-the-art segmentation and detection models, which are widely used and therefore quite relevant to the field.
(4) The results are useful and have interesting, perhaps surprising components. Namely:
(a) nnU-Net is still shown to be quite powerful, and is the best performing segmentation model, perhaps unsurprisingly given its ubiquitous success for biomedical image segmentation.
(b) However, while nnU-Net is the best performing segmentation model, it has a much higher FP rate than the other segmentation models. The author's focus on this finding and its limitations in clinical settings is a useful contribution, as is their coarse-resolution modification solution to this which resulted in much lower FP rate, but only slightly lower sensitivity drop.
(c) the "tradeoff" effect of the nnDetection thresholds between sensitivity and FP rate makes sense, but it's still a useful result.
(d) Figure 2 provides useful visualizations of the overall findings, and a "map" of performance across different sites and different models. In particular, shown on the left plot, there is typically varying ranging model performance (sensitivity and FP count) between sites for the same model, indicating that while the choice of model certainly matters, issues of domain shift between sites are quite high as well. However, we do see that the performance of the same model on different sites does "cluster" somewhat, such that there is an expected performance of a given model type in a sense. It also shows a general correlation of sensitivity with FP count.
(e) Another important contribution of this study is highlighting the challenge of trading sensitivity for specificity. While the focus in detection method papers can sometimes be on sensitivity, the latter is quite important to consider for clinical use.
(5) The paper is written well and clearly, and the presentation is polished (besides a few minor typos).

All of these factors combine to make this paper contribute a very useful performance benchmark for this task.

**Weaknesses:**

Given the reasons I discuss under "Strengths", I honestly don't think that this paper has any major weaknesses. However, I would appreciate the authors addressing some of the discussion points I mention under "Questions to address in rebuttal". Additionally, the paper would benefit from a thorough proofread, as there are a few minor typos. For example: "a a cosine annealing" at the last line of pg. 3, and "re-implentation" at the first line of the second paragraph of pg. 4.

**Detailed Comments:**

Please see my questions below.

**Justification Of The Final Rating:**

The reviewers sufficiently addressed the comments that I raised, and did fairly extensive revisions to respond to all reviewers’ comments. I appreciate reviewer B6Zn asking a few clarifying questions (particularly important for an experimental study), and I feel that the authors properly addressed them. I disagree with reviewer UGYw that the lack of technical novelty warrants a reject (especially a strong reject), as technical novelty is not necessarily required for MIDL, and I believe the contributions of this paper to be strong as an experimental study, with potential interest to the community. As such, I maintain my rating.

**Justification Of The Preliminary Rating:**

Given the paper's useful results and benchmarking, extensive datasets and modern, relevant models, and seemingly airtight experimental design, I don't think it has any major weaknesses. This submission provides an extensive experimental benchmark for a challenging task which would be of interest to the MIDL audience, which also includes interesting sub-findings which underscore the importance of analyzing deep models' potential tendencies for high false-positive rates, as well as a potential solution to this problem which could help facilitate eventual clinical adoption. Finally, this paper serves as a good example of multi-site model benchmarking in medical image analysis, a type of study which other tasks would benefit from being conducted on. I would be interested to see if some of the authors' aforementioned general findings are also present in other modalities and tasks. Altogether, I think that it poses to be a strong contribution to MIDL.

**Questions To Address In The Rebuttal:**

(1) Can the authors discuss their thoughts behind why the coarse-resolution change to nnU-Net produced such a noticeable drop in the false positive rate? Could it be because the lower resolution filters out fine-grained details which are erroneously being used by the model to make FP predictions?
(2) The authors somewhat discussed the finding that I mentioned under Strengths (4)(d) (general correlation of sensitivity and FP rate across different models and sites), but I think further elaboration on this, its implications, and potential causes, would be of interest to readers.

**Special Issue:**

Yes

---

> ### Author Response · Authors · 2025-03-07
>
> Dear reviewer, we thank you for your valuable comments and feedback and addressed it in our draft as follows:
>
> > Additionally, the paper would benefit from a thorough proofread, as there are a few minor typos.
>
> We gave the paper a thorough proof reading and hopefully caught all the typos.
>
> ---
>
> > (1) Can the authors discuss their thoughts behind why the coarse-resolution change to nnU-Net produced such a noticeable drop in the false positive rate? Could it be because the lower resolution filters out fine-grained details which are erroneously being used by the model to make FP predictions?
>
> We hypothesize the same reason as you and added it to the discussion section.
>
> ---
>
> > (2) The authors somewhat discussed the finding that I mentioned under Strengths (4)(d) (general correlation of sensitivity and FP rate across different models and sites), but I think further elaboration on this, its implications, and potential causes, would be of interest to readers.
>
> We extended the discussion section to further elaborate on this.

---

> > ### Comment · Reviewer_DVGD · 2025-03-10
> >
> > Thank you for addressing my remaining concerns. I still feel that despite the lack of technical novelty focused on by other reviewers, this evaluation study is of high value to the MIDL community, so I maintain my rating.

---

### Official Review · Reviewer_B6Zn · 2025-02-22

**Confidence:** 4
**Preliminary Rating:** 3
**Final Rating:** 4

**Summary:**

The authors propose a broad study of detection and segmentation algorithms applied on Breast MRI. The studied algorithms include several flavors of nnU-Net, SwinUNETR-V2, and nnDetection.
The evaluation is performed on the large multi-site, multi-vendor dataset.

**Strengths:**

The proposed study is well structured and offers a good picture of the performances of studied methods. The choices of the algorithms appears to be relevant.

Overall the paper is well written and reads well.

**Weaknesses:**

The paper is an evaluation study, hence has limited technical contribution. Therefore, to be enable the interest of community, more details on evaluation methodology (e.g., inputs, architecture specifics) are expected.

**Detailed Comments:**

In section 3 the authors report performances of Adapted and Coarse nnUnet, however in the text of the manuscript the adapted and coarse flavors appear to be the same. Could the authors comment?
Moreover, could the authors more clearly describe the specificities of the proposed coarse/adapted nnUnet?

In Section 2.2 the authors present the input of the models as 3D imaging, in Figure 1 the authors report the performances on Maximum Intensity Projection (MIP) images. It is not perfectly clear how the models are trained, i.e., in 2D or 3D manner?
More generally, could the authors provide more details how the images were pre-processed and fed to the networks?

It would be helful if the authors could illustrate better data from different sites to illustrate the visual differences. Could the authors do this? Maybe revising the Figure 1?

Could the authors provide more details on Figure 2 for clarity? In particular state what does the star mean?

The authors state that the annotations were drawn on public ACRIN dataset. Could the authors state whether the annotations would be released?

**Justification Of The Final Rating:**

The accomplished work and the effort by the authors is appreciated. The solid experiments may allow for a fruitful discussion with the peers. Hence, I'm ready to switch to "weak accept" rating. I still suggest the authors to revise discussions and conclusions.

**Justification Of The Preliminary Rating:**

The paper may be of interest to the community as studies several methods of segmentation and detection using Breast MRI as use case. However, the some details are expected to improve the value of the content.

**Questions To Address In The Rebuttal:**

More details about the overall evaluation (preprocessing, inputs, architecture specifics) are expected.

---

> ### Author Response · Authors · 2025-03-07
>
> Dear reviewer, we thank you for your valuable comments and feedback and addressed it in our draft as follows:
>
> > In section 3 the authors report performances of Adapted and Coarse nnUnet, however in the text of the manuscript the adapted and coarse flavors appear to be the same. Could the authors comment? Moreover, could the authors more clearly describe the specificities of the proposed coarse/adapted nnUnet?
>
> We clarified the differences in section 2.2 and added Table 4 to the appendix, which provides an overview about the differences between the different models.
>
> ---
>
> > In Section 2.2 the authors present the input of the models as 3D imaging, in Figure 1 the authors report the performances on Maximum Intensity Projection (MIP) images. It is not perfectly clear how the models are trained, i.e., in 2D or 3D manner? More generally, could the authors provide more details how the images were pre-processed and fed to the networks?
>
> We added a sentence to the first paragraph of section 2.2 to clarify that we train in 3D and added a comment to Figure 1, on why MIPs are shown, to prevent it from being confusing.
>
> The different preprocessing steps for the input images are now summarized in Table 4 in the appendix. Please let us know, if you feel that a central information is missing.
>
> ---
>
> > It would be helful if the authors could illustrate better data from different sites to illustrate the visual differences. Could the authors do this? Maybe revising the Figure 1?
>
> We agree that it would be very helpful to have an overview of the visual differences. Therefore we added Figure 3 to the appendix, which shows an example slice (precontrast, postcontrast and subtraction image) for each clinical site.
>
> ---
>
> > Could the authors provide more details on Figure 2 for clarity? In particular state what does the star mean?
>
> We added a comment to the caption if Figure 2, that clarifies that the star shows the mean over the model / site wise results. Please inform us, if you feel like other important details are missing.
>
> ---
>
> > The authors state that the annotations were drawn on public ACRIN dataset. Could the authors state whether the annotations would be released?
>
> We don't plan to release these annotations. However, we would like to point you towards the MAMA-MIA project (https://github.com/LidiaGarrucho/MAMA-MIA). In this project, tumor segmentations for the I-SPY 2 trial have been created and published, which is a superset of the ACRIN 6698 data.

---

> > ### Comment · Reviewer_B6Zn · 2025-03-15
> >
> > I would like to thank the authors for the accomplished work. The substantial amount of modifications is appreciated.
> >
> > I note that the novelty of the paper is still limited and the take-home message is not very clear: that is, the reader may hope for a more generalized message with suggested recipes eventually applicable to other organs.
> >
> > Nevertheless I believe the accomplished work may worth acceptance and the solid experiences may allow for fruitful discussions.

---

### Author Rebuttal · Authors · 2025-03-07

**Rebuttal:**

We thank all reviewers for their valuable comments and feedback. Attached you can find the revised submission and the document highlighting all changes we made.

**Supporting Material:**

/attachment/9928619e32430ffdfd0e7f7f498d7716db34b75c.zip

---

### Comment · Area_Chair_KthS · 2025-03-10
**Discussion Period**

This is a reminder for reviewers and the authors to engage in the discussion during this period.

Also, all reviewers are expected to make their final recommendations by 14th March.

Thank you for your time.

---

### Meta-Review · Area_Chair_KthS · 2025-03-16

**Recommendation:** Accept (Poster)
**Confidence:** 4

**Metareview:**

The paper proposes an extensive experimental study of lesion detection in breast MRI from a variety of datasets (sourced from seven different countries) using five different network architectures. Out of three reviewers, one recommended weak accept (updated from borderline after rebuttal), one recommended strong accept (both before and after rebuttal), and one recommended strong reject (before and after rebuttal). The reviewer who recommended strong accept clearly stated all the interesting insights provided by this experimental study. Similarly, the reviewer who recommended weak accept also acknowledges the clarity of the designed experimental study. The last reviewer, with a strong rejection, was not happy about the methodological development; however, considering that the selected paper type is a validation or application study and the emphasis of MIDL on such study, I am recommending acceptance of the paper. I encourage the authors to take into account suggestions made by the reviewers while finalizing the camera-ready version of the paper (specifically improving the discussion and conclusion of the paper).